# DIVE: Towards Descriptive and Diverse Visual Commonsense Generation

**Jun-Hyung Park**[1*]   **Hyuntae Park**[2*]   **Youjin Kang**[3]   **Eojin Jeon**[2]   **SangKeun Lee**[2,3]

[1]BK21 FOUR R&E Center for Artificial Intelligence, Korea University, Seoul, Republic of Korea
[2]Department of Artificial Intelligence, Korea University, Seoul, Republic of Korea
[3]Department of Computer Science and Engineering, Korea University, Seoul, Republic of Korea
{irish07, pht0639, yjkang10, skdlcm456, yalphy}@korea.ac.kr

## Abstract

Towards human-level visual understanding, visual commonsense generation has been introduced to generate commonsense inferences beyond images. However, current research on visual commonsense generation has overlooked an important human cognitive ability: generating descriptive and diverse inferences. In this work, we propose a novel visual commonsense generation framework, called DIVE, which aims to improve the descriptiveness and diversity of generated inferences. DIVE involves two methods, generic inference filtering and contrastive retrieval learning, which address the limitations of existing visual commonsense resources and training objectives. Experimental results verify that DIVE outperforms state-of-the-art models for visual commonsense generation in terms of both descriptiveness and diversity, while showing a superior quality in generating unique and novel inferences. Notably, DIVE achieves human-level descriptiveness and diversity on Visual Commonsense Graphs. Furthermore, human evaluations confirm that DIVE aligns closely with human judgments on descriptiveness and diversity[1].

## 1 Introduction

Humans possess a cognitive ability to reason about the rich and complex stories beyond a given visual scene, based on their background commonsense knowledge. Visual commonsense reasoning is a key to this cognition-level visual understanding (Zellers et al., 2019), which helps humans comprehend the interactions around them. As research towards the human-level visual understanding of machines, visual commonsense generation (Park et al., 2020) has been introduced. This challenging task aims to generate textual commonsense inferences about potential antecedents and consequences, as

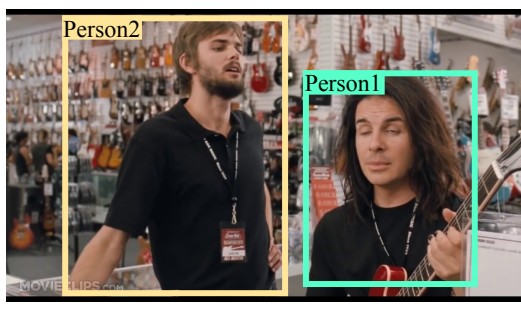

(Event) Person2 stands beside Person1 and listens intently
(Place) in a music store

(Type) Before, Person2 needed to ...

**Existing model**
"walk up to Person1"
"have a conversation"
"stand behind Person1"

**DIVE**
"meet Person1 in the *music store*"
"hear Person1 *play guitar*"
"begin talking with Person1 about *music*"

**Human**
"work in the *music store* with Person1"
"be interested in Person1's *guitar playing*"
"see Person1 *playing*"
"listen to the *music*"

Figure 1: Comparison of commonsense inferences from models and humans. Blue words represent key details.

well as the present intents of characters. Recent works on visual commonsense generation (Park et al., 2020; Xing et al., 2021) have progressed to develop vision-language models capable of generating more plausible and relevant inferences.

Despite considerable efforts in visual commonsense generation, an important aspect of humans' innate cognitive ability has been overlooked in previous studies: humans can make descriptive and diverse inferences by capturing important, specific, and detailed information within a visual scene. This ability is necessary for making precise and informative inferences about various possible scenarios in the world, but it is lacking in existing models. Figure 1 illustrates a case where model-generated inferences still fall short of human-written infer-

---

[*]These authors contributed equally to this work.
[1]Our code and dataset are available at https://github.com/Park-ing-lot/DIVE.

ences in terms of descriptiveness and diversity. Existing models often ignore key details in a given scene, leading to the generation of similar and generic inferences such as "*walk up to Person1*" and "*stand behind Person1*" that could happen in most contexts and provide minimal specific detail. In contrast, humans can create more descriptive and diverse inferences like "*work in the music store with Person1*" by considering the image's details, such as many instruments displayed behind and two men wearing employee ID cards.

We observe that this deficiency can be largely attributed to the skewed distribution of visual commonsense resources. Such resources, typically crowd-sourced, often involve many generic inferences as labels, because humans may not use detailed information from a given visual scene when annotating (Berg et al., 2012; Hessel et al., 2022). For example, more than 60% of images in Visual Commonsense Graphs (VCG) (Park et al., 2020) involve generic inferences as labels[2]. As models repeatedly learn these generic inferences, they tend to generate inferences that vaguely describe a situation, failing to capture detailed information that specifies a given scene. This limitation restricts a deeper understanding of visual information by existing vision-language models.

In this paper, we introduce DIVE (*Descriptive and dIverse Visual commonsense gEneration*), a novel visual commonsense generation framework for improving the descriptiveness and diversity of commonsense inferences generated by vision-language models. Firstly, we construct a balanced visual commonsense graph from VCG, using a carefully designed filtering method that removes generic inferences, with a focus on the semantic concentration of images utilizing the representations of CLIP (Radford et al., 2021). Furthermore, we propose a new contrastive retrieval learning method that facilitates a model to recognize specific details of an image. To verify the efficacy of DIVE, we conduct experiments on VCG (Park et al., 2020) with popular generative models (Lewis et al., 2020; Li et al., 2022b). Our experiments verify that DIVE generates more descriptive and diverse inferences compared with state-of-the-art visual commonsense generation models. Notably, DIVE achieves human-level descriptiveness and diversity scores on VCG, significantly improving the

generation quality of unique and novel inferences. We further conduct human evaluations on the plausibility, descriptiveness, and diversity of generated inferences, which confirm that DIVE aligns closely with humans' judgment of descriptiveness and diversity.

Our main contributions are as follows,

- We propose a novel framework for visual commonsense generation, called DIVE, which enhances the capability of vision-language models to generate descriptive and diverse inferences about visual scenes. To the best of our knowledge, this is the first work to address descriptiveness and diversity in visual commonsense generation, providing deep insights.

- We develop generic inference filtering and contrastive retrieval learning methods to facilitate descriptive and diverse visual commonsense generation of vision-language models on the skewed visual commonsense resources.

- Our extensive experiments verify that DIVE outperforms state-of-the-art visual commonsense generation models in terms of the descriptiveness and diversity on VCG.

## 2   Related Work

**Visual commonsense reasoning.**   With the goal of reasoning beyond visual recognition, the community has actively explored several visual commonsense reasoning tasks. Zellers et al. (2019) have proposed a visual commonsense reasoning benchmark to test if a model can identify an answer with rationale, given a question that requires a thorough understanding of images based on commonsense knowledge. Hessel et al. (2022) have proposed an abductive reasoning benchmark beyond literal image contents to evaluate the capacity of models to retrieve relevant inferences, localize evidence, and compare plausible inferences. Li et al. (2022a) have introduced a video QA benchmark that requires understanding of evidences and commonsense reasoning over time. Yu et al. (2022) have proposed an audiovisual commonsense reasoning benchmark focusing on physical knowledge, which requires an understanding on multi-sensory inputs. However, these benchmarks evaluate models in a question answering format, limiting the evaluation of the models' capability to generate commonsense inferences. To address this, Park et al. (2020) have

---

[2]In VCG, 61% of images involve the 100 most frequent inference results as their labels, which are predominantly generic, like "talk to Person1" and "eat dinner".

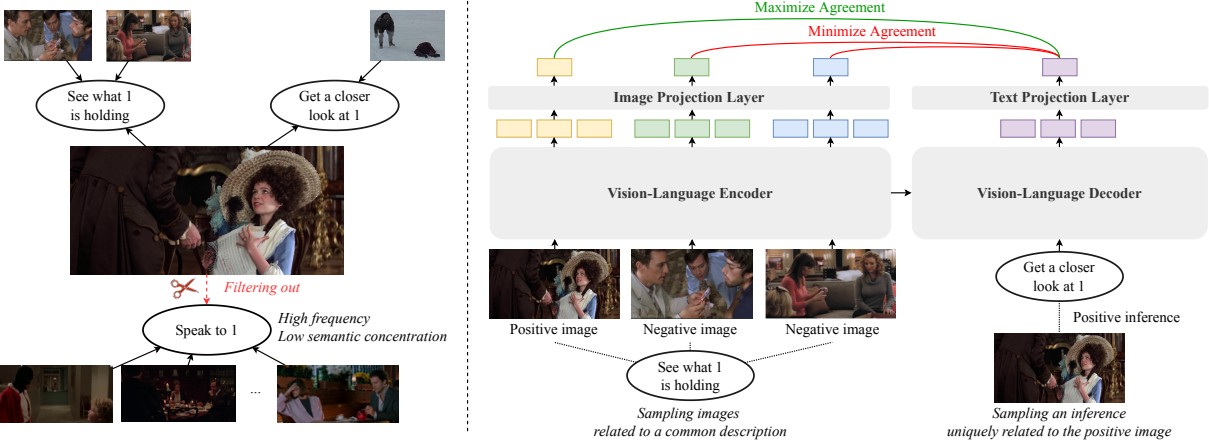

|  (a) Generic Inference Filtering  |  (b) Contrastive Retrieval Learning  |

Figure 2: Illustration of DIVE. (a) Generic Inference Filtering: Filtering out inferences with high frequency and low semantic concentration of related images. (b) Contrastive Retrieval Learning: Learning to maximize the agreement between a pair of an image and its unique corresponding inference. Events and places are omitted for clarity.

introduced a visual commonsense generation task with VCG, which asks models to generate inferences about potential antecedents, consequences, and the intents of people based on images and their corresponding textual descriptions. This requires models to generate free-form textual descriptions of inference results, which is more challenging and in line with the process of humans reasoning (Jung et al., 2022). As an approach to visual commonsense generation, Park et al. (2020) have extended pre-trained language models like GPT-2 (Radford et al., 2019) by fine-tuning them to process visual information. KM-BART (Xing et al., 2021) has constructed vision-language datasets for continual pre-training of BART (Lewis et al., 2020), aiming to enhance its commonsense knowledge with additional data. Recent studies have proposed large-scale pre-trained vision-language models (Li et al., 2022b; Wang et al., 2022; Lu et al., 2023; Han et al., 2023), which have shown promising results across various vision-language tasks. Several studies have extended the modalities of vision-language transformers to process audio and time information for a holistic understanding from all our senses (Zellers et al., 2022; Zong and Sun, 2023). In this work, based on VCG (Park et al., 2020) and vision-language models (Xing et al., 2021; Li et al., 2022b), we propose a novel framework focusing on descriptive and diverse commonsense generation, which is important but overlooked in prior research on visual commonsense reasoning.

**Descriptive and diverse text generation.** Generating meaningful and informative text has been a challenging goal in many NLP fields including response generation, image captioning, and scene graph generation (Li et al., 2016; Dai and Lin, 2017; Zhao et al., 2017; Jiang and de Rijke, 2018; Luo et al., 2018; Wu et al., 2020; Tang et al., 2020). In these works, several problems have been identified in current text generation frameworks. First, generation models often prefer "safe" output sentences that use very high-frequency expressions, and they tend to describe only the obvious facts while ignoring key details (Li et al., 2016). Second, the standard training objective function for text generation can promote the generation of generic text without diversity (Li et al., 2016). Third, conventional metrics for text generation, such as BLEU and CIDEr, tend to give high scores to inferences with very common n-grams, as they prioritize n-gram overlap (Liu et al., 2019). Thus, as the first exploration of descriptive and diverse commonsense inferences, we propose filtering and training methods to prevent models from favoring "safe" sentences. Additionally, we construct datasets for evaluation, introduce various metrics, and conduct human evaluations to particularly evaluate descriptiveness and diversity of generated inferences.

## 3 Methodology

In this section, we introduce DIVE, a framework designed for descriptive and diverse commonsense generation. First, we propose a generic inference filtering method that balances the distribution of

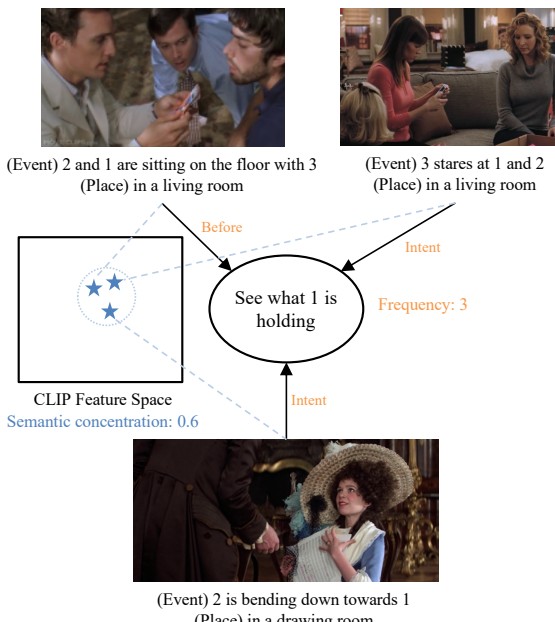

(Event) 2 and 1 are sitting on the floor with 3 (Place) in a living room

(Event) 3 stares at 1 and 2 (Place) in a living room

Before

Intent

See what 1 is holding

Frequency: 3

CLIP Feature Space
Semantic concentration: 0.6

Intent

(Event) 2 is bending down towards 1 (Place) in a drawing room

Figure 3: Illustration of measuring the semantic concentration and frequency of inferences.

visual commonsense resources (Section 3.1). Then, we propose a contrastive retrieval learning method that facilitates vision-language models to identify information specific to a given image (Section 3.2). Figure 2 illustrates the overall procedure of DIVE.

## 3.1 Generic Inference Filtering

Since the skewed distribution of VCG can cause vision-language models to favor generic inferences, we construct a balanced VCG based on our generic inference filtering method. We identify generic inferences based on their frequencies and how their related images are semantically concentrated, because a generic inference is expected to be frequent and associated with a broader range of images.

Given a visual commonsense graph $G = (I, E, P, C, R)$, where $I$, $E$, $P$, and $C$ denote sets of images, events, places, and commonsense descriptions, respectively, $R$ is the set of edges in the graph comprising visual commonsense inferences $R_{ij} = (I_i, E_i, P_i, r, C_j)$. Here $I_i \in I, E_i \in E, P_i \in P, C_j \in C$, and a reasoning type $r \in \{\text{before}, \text{after}, \text{intent}\}$. Then, we measure the semantic concentration of images related to a commonsense description. Specifically, we measure the average cosine similarity of feature representations of the related images as follows:

$$S(C_j) = \frac{\sum_{x \in \mathcal{G}(C_j)} \sum_{y \in \mathcal{G}(C_j)} \text{sim}(\mathcal{F}(x), \mathcal{F}(y))}{|\mathcal{G}(C_j)|^2},$$
(1)

where $\mathcal{F}(\cdot)$ represents an image encoder, $\mathcal{G}(C_j)$ is the set of images related to a commonsense description $C_j$. When calculating the similarity, we utilize the average of feature representations in the final hidden layer of CLIP (Radford et al., 2021). The measured value indicates how closely the related images lie in the feature space, that is, how specific an inference is.

Using this semantic concentration $S(C_j)$ and the frequency $|\mathcal{G}(C_j)|$ of a commonsense description $C_j$, we identify and filter out generic inferences for each commonsense description. Figure 3 illustrates an example to measure the semantic concentration and frequency on visual commonsense graphs. Inspired by the frequency-based filtering method for words (Mikolov et al., 2013), we calculate the filtering probability $P_f(C_j)$ for a commonsense description $C_j$, which is defined as follows:

$$P_f(C_j) = 1 - \sqrt{\frac{t \times S(C_j)}{|\mathcal{G}(C_j)|}},$$
(2)

where $t$ is a threshold. Finally, we deterministically filter $\lfloor P_f(C_j)|\mathcal{G}(C_j)| \rfloor$ inferences between $C_j$ and related images with the lowest average similarity to the others.

## 3.2 Contrastive Retrieval Learning

Although generic inference filtering can effectively reduce the skewness of the distribution, we observe that over-filtering negatively impacts the quality of models' inference results, primarily due to the reduction in the number of training examples. Additionally, it has been noted that a specifically designed training objective function to improve descriptiveness and diversity is beneficial, as the standard generative objective function may lead to generic generation (Li et al., 2016; Luo et al., 2018). These two observations underscore the need to develop novel training methods that improve descriptiveness and diversity in visual commonsense generation, used in conjunction with the filtering method.

Here we propose a new contrastive retrieval learning method that encourages models to generate descriptive and diverse inferences about an image. The key motivation of our method is that models need to recognize detailed objects and interactions within images to generate descriptive and diverse inferences. Our method trains a model to retrieve the original image from which a given inference is derived, within a set of similar images,

in a contrastive manner. This approach is expected to facilitate models to identify the differences in detailed objects and interactions among similar images, thereby aligning with our motivation.

First, we construct a set of similar images $H \subset \mathcal{G}(C_j)$ that share the same commonsense description $C_j$ as an inference. To identify similar images, we sample images that share the same inference result, based on our intuition that they have semantic similarity that leads to the same inference result. Subsequently, we identify a pair of an image and its corresponding commonsense description $(h_p, s_p)$, where $h_p \in H$ and $s_p$ is a commonsense description uniquely related to $h_p$ among the images in $H$. We consider $h_p$ as a positive image and $s_p$ as a positive inference, while treating the other images $h_k \in H$ as negative images. We then define an agreement function based on cosine similarity as follows:

$$\sigma(h, s) = \exp(\text{sim}(V_h, T_s)), \quad (3)$$

where $V_h$ and $T_s$ denote the feature representations of an image $h$ and a text $s$ from a vision-language model being trained, which are extracted by averaging the output feature vectors from the final layers of the encoder and decoder, respectively. It is worth noting that we obtain the image and text representations by averaging the projected representations from the image encoder and text decoder of a vision-language model, respectively. Finally, we define our contrastive retrieval loss as follows:

$$\mathcal{L}_{crl}(h_p, s_p, H) = -\log \frac{\sigma(h_p, s_p)}{\sum_{i=1}^{|H|} \sigma(h_i, s_p)}. \quad (4)$$

Based on the proposed method, we aim to train models to consider unique components that lead to specific inference results, rather than common components that lead to more generic inference results shared by multiple images.

We integrate our contrastive retrieval loss with the original language modeling loss of visual commonsense generation (Park et al., 2020). For a given image $h \in H$, we can identify corresponding ground-truth inference $s = \{w_1, w_2, ..., w_k\}$ as a sequence of tokens. Then, the original loss is defined as follows:

$$\mathcal{L}_{org}(h, s) = -\sum_{i=1}^{|s|} \log P(w_i|w_{<i}, h). \quad (5)$$

The final objective function is as follows:

$$\mathcal{L}(h_p, s_p, H) = \mathcal{L}_{org}(h_p, s_p) + \lambda \mathcal{L}_{crl}(h_p, s_p, H), \quad (6)$$

where $\lambda$ is a non-negative hyper-parameter for balancing the objective functions. It is noteworthy that we randomly select one inference for the loss calculation if multiple inferences are associated with an image $h_p$. We construct $H$ for each example in a batch and if a uniquely related commonsense description does not exist, we exclude the contrastive loss.

## 4 Experiments

In this section, we demonstrate the effectiveness of DIVE by comparing it with existing methods.

### 4.1 Experimental Setup

**Dataset.** We conduct the experiments on the VCG dataset (Park et al., 2020), which is a large-scale visual commonsense graph. We train models with DIVE on the filtered VCG training set. In addition, we evaluate models on the original VCG validation set, the unique VCG validation set, and the novel VCG validation set. The unique VCG validation set is a subset of the original set that consists of inferences with commonsense descriptions that appear once in the original set. The novel VCG validation set is a subset of the original set that consists of inferences with commonsense descriptions that do not appear in the training set. We expect that the unique and novel subsets predominantly contain specific inferences, since they exclude duplicate examples. For both subsets, we discard the inferences of images with fewer than five commonsense descriptions. The statistics of the dataset are reported in Appendix A.

**Baselines.** We mainly compare our results with those of VisualCOMET (Park et al., 2020), KM-BART (Xing et al., 2021), and BLIP (Li et al., 2022b). VisualCOMET (Park et al., 2020) extends a pre-trained GPT-2 model (Radford et al., 2019) with 126 million parameters to incorporate visual and textual information. KM-BART is based on a pre-trained BART-base model (Lewis et al., 2020) with 141 million parameters and conducts additional pre-training with image captioning data. BLIP (Li et al., 2022b) is a pre-trained generative vision-language transformer with 247 million parameters. All the baselines are fine-tuned on VCG following the fine-tuning settings specified in their original papers.

**Implementation details.** We fine-tune both vision-language BART and BLIP on VCG using

| Model | Length | Yngve | Dist-2 | Dist-3 | R@1 | R@5 | R@10 | Entropy | Unique | Novel |
|---|---|---|---|---|---|---|---|---|---|---|
| VisualCOMET | 4.733 | 7.68 | 58K | 127K | 29.56 | 53.76 | 64.38 | 19.38 | 42.28 | 45.24 |
| KM-BART | 4.614 | 7.37 | 67K | 159K | 37.38 | 62.03 | 71.75 | 18.76 | 57.61 | 38.57 |
| BLIP | 4.659 | 7.50 | 77K | 174K | 66.21 | 88.52 | 93.52 | 18.56 | 58.48 | 40.82 |
| $DIVE_{BART}$ (ours) | 5.156 | **8.88** | 84K | 207K | 51.40 | 77.47 | 85.02 | **21.09** | **76.09** | 54.20 |
| $DIVE_{BLIP}$ (ours) | **5.223** | 8.80 | **93K** | **221K** | **77.14** | **94.78** | **97.38** | 20.91 | 76.05 | **56.50** |
| Human | 4.858 | 8.15 | 93K | 190K | - | - | - | 20.71 | 74.34 | 54.98 |

Table 1: Evaluation of descriptiveness and diversity on the original VCG validation set.

| Model | Length | Yngve | Dist-2 | Dist-3 | R@1 | R@5 | R@10 | Entropy | Unique | Novel |
|---|---|---|---|---|---|---|---|---|---|---|
| VisualCOMET | 4.811 | 8.20 | 7.1K | 10.1K | 34.65 | 57.27 | 67.12 | 19.20 | 79.33 | 48.63 |
| KM-BART | 4.638 | 7.44 | 8.3K | 11.9K | 23.43 | 60.19 | 69.56 | 18.83 | 91.33 | 42.67 |
| BLIP | 4.646 | 7.38 | 8.2K | 11.7K | 66.74 | 87.42 | 92.03 | 18.43 | 89.44 | 43.45 |
| $DIVE_{BART}$ (ours) | **5.169** | **8.94** | **10.1K** | 14.9K | 31.78 | 74.55 | 82.36 | **21.09** | **96.88** | 56.74 |
| $DIVE_{BLIP}$ (ours) | 5.098 | 8.82 | 10.1K | **15.0K** | **77.01** | **94.94** | **97.17** | 20.93 | 95.34 | **57.33** |
| Human | 5.792 | 10.39 | 14.9K | 21.3K | - | - | - | 26.11 | 100.0 | 100.0 |

Table 2: Evaluation of descriptiveness and diversity on the unique VCG validation set.

| Model | Length | Yngve | Dist-2 | Dist-3 | R@1 | R@5 | R@10 | Entropy | Unique | Novel |
|---|---|---|---|---|---|---|---|---|---|---|
| VisualCOMET | 4.788 | 8.26 | 4.6K | 5.8K | 46.46 | 69.77 | 78.79 | 19.52 | 71.55 | 45.75 |
| KM-BART | 4.637 | 7.41 | 5.1K | 6.6K | 32.18 | 70.37 | 78.37 | 19.00 | 86.45 | 40.47 |
| BLIP | 4.614 | 7.47 | 5.0K | 6.4K | 72.93 | 92.18 | 95.63 | 18.84 | 84.61 | 41.20 |
| $DIVE_{BART}$ (ours) | 5.165 | **8.85** | **6.0K** | **8.1K** | 38.64 | 81.80 | 89.32 | **21.24** | **97.77** | 59.12 |
| $DIVE_{BLIP}$ (ours) | **5.186** | 8.80 | 6.0K | 8.1K | **85.86** | **97.29** | **98.42** | 21.23 | 95.48 | **59.69** |
| Human | 5.515 | 11.32 | 9.5K | 12.5K | - | - | - | 24.44 | 100.0 | 100.0 |

Table 3: Evaluation of descriptiveness and diversity on the novel VCG validation set.

our DIVE framework. DIVE with vision-language BART ($DIVE_{BART}$) is based on the BART architecture (Lewis et al., 2020) with several modifications to process vision-language inputs consistent with KM-BART (Xing et al., 2021). We use projected Region of Interest (RoI) feature representations of an image from the pre-trained Faster R-CNN (Ren et al., 2015) as a vision input to vision-language BART. For each image, Faster R-CNN detects several objects and generates their bounding boxes and classification results as RoI features. We extract the feature representations fed into the final classification layer of Faster R-CNN. The extracted representations are subsequently projected to fit the embedding dimension of the language models. DIVE with BLIP ($DIVE_{BLIP}$) uses the BLIP architecture (Li et al., 2022b) while keeping the visual encoder layers frozen. We use the input formats consistent with Xing et al. (2021). We set the filtering threshold to 10 and extract two sampled images from each batch example for contrastive retrieval learning. We generate five inferences per example using nucleus sampling (Holtzman et al., 2020) with $p = 0.9$. All our experiments are conducted on six NVIDIA RTX A6000 GPUs. Detailed training hyper-parameters are specified in Appendix E.

**Metrics.** We employ a variety of automatic evaluation metrics following previous works (Liu et al., 2019; Park et al., 2020) to evaluate the descriptiveness and diversity. We evaluate the sentence length (Length), syntactic complexity (Yngve (Yngve, 1960)), number of distinct n-grams in the whole generated inferences (Dist-2 and Dist-3 (Xu et al., 2018)), image retrieval performance given the generated inference (R@1, R@5, and R@10 (Liu et al., 2019)), word-level entropy that measures the average log probability of the uni-gram words of a generated inference in the training set (Entropy (Mou et al., 2016)), percentage of unique inferences within the generated inferences (Unique (Park et al., 2020)), and percentage of novel inferences that are not seen in the training set (Novel (Park et al., 2020)). A set of more descriptive and diverse inferences will show higher scores in terms of the above-mentioned metrics. We additionally employ conventional metrics to evaluate the quality of generated inferences including BLEU (Papineni et al., 2002), METEOR (Denkowski and Lavie, 2014), CIDEr (Vedantam et al., 2015), and SPICE (Anderson et al., 2016). However, conventional metrics are limited to evaluate accuracy of descriptive and diverse inferences (Li et al., 2016), because they give a high score to the inferences that

| Model | B-2 | M | C | S |
|---|---|---|---|---|
| VisualCOMET | 13.50 | 11.37 | 18.28 | 5.53 |
| KM-BART | **14.27** | 11.22 | **21.22** | 6.85 |
| BLIP | 13.93 | 11.33 | 20.81 | 6.90 |
| $DIVE_{BART}$ (ours) | 13.33 | **11.48** | 20.26 | **7.33** |
| $DIVE_{BLIP}$ (ours) | 12.86 | 11.39 | 19.07 | 6.90 |

Table 4: Evaluation of generation quality on the original VCG validation set.

| Model | B-2 | M | C | S |
|---|---|---|---|---|
| VisualCOMET | 18.16 | 11.94 | 18.11 | 5.50 |
| KM-BART | 17.80 | 11.26 | 19.65 | 7.36 |
| BLIP | 16.69 | 11.11 | 17.92 | 7.03 |
| $DIVE_{BART}$ (ours) | **18.83** | **12.52** | **21.20** | **8.08** |
| $DIVE_{BLIP}$ (ours) | 17.96 | 12.32 | 19.70 | 7.45 |

Table 5: Evaluation of generation quality on the unique VCG validation set.

| Model | B-2 | M | C | S |
|---|---|---|---|---|
| VisualCOMET | 18.07 | 12.23 | 16.78 | 5.34 |
| KM-BART | 18.41 | 11.70 | 19.43 | 7.13 |
| BLIP | 17.15 | 11.66 | 17.66 | 6.82 |
| $DIVE_{BART}$ (ours) | **19.04** | **12.23** | **22.52** | **8.36** |
| $DIVE_{BLIP}$ (ours) | 18.01 | 11.91 | 20.17 | 7.54 |

Table 6: Evaluation of generation quality on the novel VCG validation set.

have many n-gram overlaps with ground-truth inferences, which are predominantly generic in visual commonsense resources. Thus, we further conduct human evaluations on the generated inferences.

## 4.2 Main Results

We first evaluate the descriptiveness and diversity in visual commonsense generation of vision-language models. In Table 1, we compare our DIVE models with state-of-the-art visual commonsense generation models on the original VCG validation set. We observe that our DIVE models outperform the baselines in all evaluation metrics for descriptiveness and diversity. Particularly, DIVE models reach human-level descriptiveness and diversity on the original VCG validation set. These results confirm that our DIVE framework effectively augments vision-language models with the capability for generating descriptive and diverse commonsense inferences, showing significant improvements over existing vision-language models. In addition, as shown in Tables 2 and 3, our DIVE models consistently outperform the baselines in terms of descriptiveness and diversity on the unique and novel VCG validation sets.

To evaluate the quality of visual commonsense generation, we compare our DIVE models with the

| $DIVE_{BART}$ vs. | Plausible | | Descriptive | | Diverse | |
|---|---|---|---|---|---|---|
| | Win | Lose | Win | Lose | Win | Lose |
| VisualCOMET | **61.7** | 38.3 | **54.7** | 45.3 | **68.9** | 31.1 |
| KM-BART | **59.8** | 40.2 | **56.0** | 44.0 | **56.7** | 43.3 |

Table 7: Human evaluation on the original VCG validation set.

baselines using conventional evaluation metrics, as shown in Tables 4-6. On the original VCG validation set, our models show a comparable quality to the baselines. However, since the results are derived from the original VCG validation set, which involves many generic inferences, these results may be affected by the limitations of the conventional metrics mentioned in Section 4.1. To more precisely evaluate the generation quality of descriptive and diverse inferences, we further conduct experiments on the unique and novel VCG validation set. In these experiments, we observe that our DIVE models exhibit significantly better generation quality than the baselines by improving the scores up to 17.3% in terms of SPICE. These results demonstrate that DIVE has a better capability for generating descriptive and diverse inferences compared with existing models.

## 4.3 Human Evaluation Results

We present human judgments on the plausibility, descriptiveness, and diversity of the generated inferences. We conduct human evaluations with workers from Amazon Mechanical Turk[3].

Following Xing et al. (2021), we generate 450 inferences with three different models, including $DIVE_{BART}$, VisualCOMET (Park et al., 2020), and KM-BART (Xing et al., 2021), for pair-wise comparison. We generate five pairs of inferences for each inference type. For each example, we construct pairs of inferences generated by DIVE and one of the baselines, and sets of all inferences generated by each model. Then, we ask three annotators to choose a better inference based on the following three metrics: 1) **plausible**: which inference seems more plausible and reasonable to an image, 2) **descriptive**: which inference explains the image more informatively and specifically, and 3) **diverse**: which set of inferences seems more diverse in meanings and expressions.

The results are shown in Table 7. For plausibility, our $DIVE_{BART}$ model outperforms the state-of-

[3]We selected annotators who achieved ≥99% on the qualification HIT before. The workers were paid proportionally at $0.15 per example, which resulted in at least $16/hr.

|           | GIF | CRL | SPICE | R@1   | Unique |
|-----------|-----|-----|-------|-------|--------|
|           | ✓   | ✓   | **7.33** | **51.40** | **76.09** |
| DIVE$_{BART}$ | ✓ | -   | 6.89  | 48.87 | 73.49  |
|           | -   | ✓   | 7.05  | 32.93 | 56.56  |
|           | -   | -   | 7.19  | 37.38 | 58.12  |

Table 8: Ablation study on the original VCG validation set. GIF and CRL denote generic inference filtering and contrastive retrieval learning, respectively.

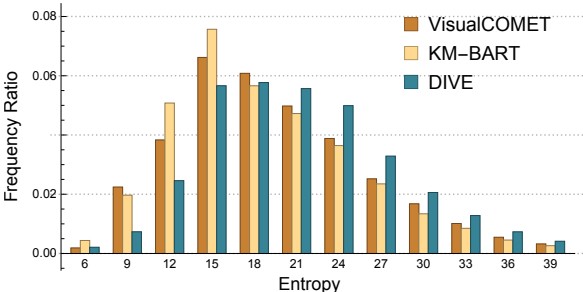

Figure 4: Distribution of generated inferences in relation to word-level entropy.

the-art visual commonsense generation models in approximately 60% of cases. This result suggests that humans generally perceive inferences generated by DIVE better than those from the baselines on the original VCG validation set, also revealing the limitations of conventional metrics for quality evaluation. For descriptiveness and diversity, we observe consistent results of winning in approximately 55% and 55% - 70%, respectively. This demonstrates that DIVE aligns more closely with human judgments on descriptiveness and diversity.

# 5 Analysis

In this section, we conduct analyses of the components and results of DIVE$_{BART}$.

## 5.1 Ablation Study

To better understand the contributions of each component in DIVE to performance improvements, we conduct ablation studies on generic inference filtering and contrastive retrieval learning. The results are shown in Table 8. We find that training models without our filtering method results in a significant degradation in the R@1 and Unique scores, which highlights that balancing the distribution of the visual commonsense resources is crucial for generating descriptive and diverse inferences. In addition, our contrastive retrieval learning method universally improves the three metrics when combined with the filtering method, showing its contributions to the improvements in generation quality, descriptiveness and diversity. Nevertheless, the contrastive retrieval learning method degrades the performance when applied alone. We speculate that this is because a wide range of images can be frequently sampled as negative ones if generic inferences are not eliminated, failing to meet the motivation of the method that trains models to recognize the detailed differences among similar images. This observation also shows that the components of DIVE are complementary to each other for the performance improvements.

## 5.2 Informativeness of Inferences

We analyze the amount of information contained in generated inferences by measuring word-level entropy (Mou et al., 2016). Figure 4 shows the distribution of generated inference on the original VCG validation set in relation to word-level entropy. The y-axis represents the ratio of the number of generated inferences for the corresponding interval of entropy in the x-axis. Each value $k$ in the x-axis represents the interval between $k - 1.0$ and $k + 1.0$. We can observe that DIVE generates inferences with relatively high entropy, which implies the improvements in their informativeness.

## 5.3 Qualitative Analysis

We present qualitative examples of DIVE compared to baselines in Figure 5. It demonstrates that DIVE can generate more descriptive and diverse inferences compared to the baselines. As can be observed from the figure, DIVE effectively generates unique and novel inferences applicable to the given situations, utilizing specific expressions related to the image, such as "disturb", "thirst", etc. In contrast, the baselines frequently generate simple, generic, and seen descriptions. Interestingly, DIVE sometimes generates more descriptive and diverse inferences compared to human annotations like the Intent inferences in Figure 5 (a). This further implies that existing automatic evaluation can underestimate the scores of DIVE due to lexical differences from human annotations.

Despite the promising results, DIVE generates some irrelevant inferences to the context even if the detailed information is explicitly given. This shows that vision language models still lack some commonsense knowledge and reasoning ability to accurately reason over recognized context.

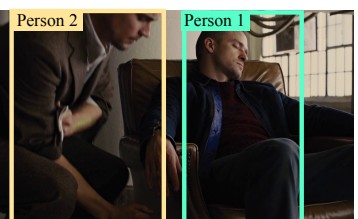

| | Before | Intent | After |
|---|---|---|---|
| Human | • be too exhausted to leave
• plop down in the chair for a nap
• take a break from studies
• relax in the chair | • rest
• take a nap | • sense someone is near him
• wake up suddenly
• be tapped by 2
• wake up and be startled |
| DIVE | • have laid down in the chair
• start to feel sick
• have taken his day off work
• feel comfortable in the chair
• realize that he was tired | • get his rest
• make 2 go away
• get the night over with
• get a good night's sleep
• get a good night's sleep | • wake up when 2 is approaching him
• roll over on his side
• be disturbed by 2's idea of action
• try to fall asleep again
• wake up refreshed |
| KM-BART | • put on sunglasses
• grab a seat
• decide to sit down on the chair
• get into a comfortable position
• put on a tie | • get some rest
• get a short nap
• get some rest
• get some rest
• get some rest | • arrive at his destination
• be woken up by 3
• fall sleep
• wake up feeling refreshed
• fall asleep |
| VisualCOMET | • lie down
• sneak in
• fall asleep
• get into the chair
• be tired | • spend time away from his family
• sneak a peek at their phone
• end the day quickly
• get some rest | • call 2 over
• took a nap
• ask 2 for a job
• receive his phone call from 2
• be questioned by 2 |

(a)

**(Event) Person1 is sleeping in a chair**

**(Place) in a loft**

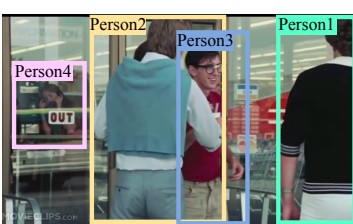

| | Before | Intent | After |
|---|---|---|---|
| Human | • be too exhausted to leave
• plop down in the chair for a nap
• take a break from studies
• relax in the chair | • drink some soda
• go to his car | • joke around with his friends
• get bullied by the other kids
• start getting bullied by 1 and 2
• get his drink poured over his head |
| DIVE | • buy the cup
• purchase a drink at the pump
• notice that 1 and 4 are acting suspicious
• pay for the cup
• enter the store and buy a drink | • carry his purchases home
• spend his break at the store
• drink his drink on the sidewalk
• stir the drink to quench the thirst of 3
• go home with their purchases | • watch 2 and 4 pass by
• make another drink
• drink his drink until it is gone
• take his purchases home
• enter a vehicle outside |
| KM-BART | • decide where to go
• wait in line in line
• buy the drink at the store
• grab a cup from the counter
• pay for his drink | • leave and go home
• go home
• leave for the day
• watch a movie in the store
• leave the store | • leave the store
• take a drink form the cup
• go home and rest
• get into the car
• buy food for his and 1's date |
| VisualCOMET | • buy alcohol
• purchase a drink
• walk into the store
• get the drink from the bar
• go into the store | • drink alcohol
• enjoy a drink
• go home
• quench this thirst
• get another drink | • talk with 1 and 2
• greet 1 and 2
• walk past 1 and 2
• enjoy a drink with 1 and 2
• go home with 1 and 2 |

(b)

**(Event) Person3 walks out of the store holding a drink cup**

**(Place) near a shop entrance**

Figure 5: Comparison of generation examples from DIVE, KM-BART (Xing et al., 2021), VisualCOMET (Park et al., 2020), and human annotations in VCG validation set. We mark red if the inference is implausible and blue if the inference is both unique and novel.

## 6 Conclusion

We have presented DIVE to improve the descriptiveness and diversity of vision-language models in visual commonsense generation. We have proposed a generic inference filtering method to balance the skewed distribution of visual commonsense resources, based on the frequency and semantic concentration of images. In addition, we have proposed a contrastive retrieval learning method to promote the descriptiveness and diversity of vision-language models, by leveraging the structural information from visual commonsense graphs. Through extensive experiments on VCG, we have verified that DIVE is capable of generating descriptive and diverse inferences about visual scenes, significantly outperforming state-of-the-art visual commonsense generation models. Particularly, our human evaluations have confirmed that DIVE indeed captures specific visual information, leading to improvements in plausibility, descriptiveness, and diversity.

## Limitations

While we have demonstrated that DIVE effectively improves the descriptiveness and diversity of generated inferences, there are some limitations that present promising avenues for future research. First, our filtering method leads to a loss of training data, which can limit the capabilities of vision-language models. We plan to investigate data augmentation methods for visual commonsense generation, such as utilizing external data (Xing et al., 2021) or data generated by foundation models (West et al., 2022). In addition, as the first work that explores the descriptiveness and diversity in visual commonsense generation, we have focused on the evaluations on VCG, which is the most representative dataset for visual commonsense generation. Nevertheless, the ability to capture detailed information from an image and to generate descriptive and diverse inferences would be significantly beneficial to various visual reasoning and generation tasks (Hessel et al., 2022; Zang et al., 2021; You et al., 2022), as well as reasoning tasks over audio and video (Yu et al., 2022; Li et al., 2022a). We thus plan to investigate the efficacy of DIVE on more diverse tasks and modalities. Finally, in Section 5.3, we have observed that DIVE occasionally generates irrelevant inferences to the given context possibly due to the lack of commonsense knowledge and reasoning ability. Future work could focus on enhancing the commonsense knowledge and reasoning ability in vision-language models (Li et al., 2022c; Han et al., 2023).

## Acknowledgements

This work was supported by the Basic Research Program through the National Research Foundation of Korea (NRF) grant funded by the Korea government (MSIT) (2021R1A2C3010430) and Institute of Information & Communications Technology Planning & Evaluation (IITP) grant funded by the Korea government (MSIT) (No.2019-0-00079, Artificial Intelligence Graduate School Program (Korea University)).

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

## Appendix

We supplement our main content with dataset analysis and additional experiments.

## A  Dataset Statistics

Table 9 presents statistics of the VCG training set, categorizing the inferences by types. It provides the number of inferences, the number of unique inferences, and the frequency of the 50 most frequent inferences, along with their ratios. It is evident that the 50 most frequent inference results, accounting for 0.1% of the total inferences, occupy a significant proportion of the entire dataset. Tables 10 and 11 show the comparison between the original and processed datasets.

| | Before | After | Intent |
|---|---|---|---|
| # Inferences | 467,025 | 469,430 | 237,608 |
| # Unique inferences | 270,419 | 282,893 | 166,466 |
| # Top-50 inferences (Ratio) | 25,103 (5.38%) | 24,894 (5.30%) | 8,012 (3.37%) |

Table 9: Detailed statistics of the VCG training set

The bar charts in Figure 6 depict the 10 most frequent inference results for each inference type, along with their frequencies in the training set. These charts obviously show an imbalance among the inference results, with "walk into the room", "talk to person", and "leave the room" being the most frequent ones. The images in VCG are derived from the Visual Commonsense Reasoning dataset (Zellers et al., 2019), which contains carefully selected images depicting social interactions between at least two people. In such context, the most frequent inferences are indeed simple and generic, applicable to most images in VCG portraying two individuals.

## B  Decoding Strategy

We primarily use nucleus sampling (Holtzman et al., 2020), a powerful decoding method, to obtain more diverse and descriptive sentences as our objective is to avoid generating generic inferences. We have also considered other options, such as greedy search, beam search, and top-k sampling, for generating commonsense descriptions. The comparison results of different decoding strategies are presented in Table 12. Except for greedy search, we generate five sentences per example.

Compared with nucleus sampling, top-k sampling shows a slightly better performance in terms

| Training set | #Image | #Inference |
|---|---|---|
| Original | 47,595 | 1,174,063 |
| Filtered | 47,595 | 949,284 |

Table 10: Statistics of VCG training sets.

| Validation set | #Image | #Inference |
|---|---|---|
| Original | 13,768 | 146,332 |
| Unique | 1,109 | 7,067 |
| Novel | 567 | 3,485 |

Table 11: Statistics of VCG validation sets.

of diversity and descriptiveness, but the quality of the generated results is poor. In addition, we observe that beam search and greedy search exhibit poor results in generating descriptive and diverse sentences. Upon evaluation, we find that nucleus sampling is closer to the optimal decoding strategy when considering all the generation quality, descriptiveness, and diversity.

## C  Similarity Evaluation Analysis

We report the results of the similarity evaluation among various design choices including image-only, text-only, and combined image-text settings in Figure 7 and 8. These figures illustrate comparisons of the top-2 similar and dissimilar images on two randomly sampled examples. We first verify that our similarity metric only using an image is not over-influenced by visual similarity. Upon analyzing images categorized by our metric as similar and dissimilar, we are unable to identify any distinct visual patterns that distinguish them. In addition, despite every image in the VCG dataset having its own accompanying textual event descriptions, involving texts in similarity evaluation (solely relying on texts or combining them with images) can lead to over-influence by overlapping words and text lengths, resulting in leaving irrelevant images or filtering out relevant, descriptive images.

## D  Number of Similar Images

In our contrastive retrieval learning method, we additionally sample similar images for one example in a batch. Specifically, for one example, we first find the images that are related to the same inference and then sample images and inference sentences uniquely related to those images. Therefore, the images that are related to at least one generic inference and one unique inference could be sampled. In our main experimental setup, we primarily sample one image for each example, given that we filter

| Model | Decoding Strategy | B-2 | C | Entropy | Unique |
|---|---|---|---|---|---|
| | Nucleus ($p = 0.9$) | 13.33 | 20.26 | 21.09 | 76.09 |
| | Nucleus ($p = 0.7$) | 15.78 | 25.05 | 19.99 | 63.06 |
| DIVE$_{BART}$ | Top-k ($k = 50$) | 12.40 | 18.82 | 21.67 | 80.82 |
| | Beam (10 beams) | 22.01 | 38.38 | 16.11 | 22.59 |
| | Greedy | 21.82 | 37.75 | 17.84 | 40.70 |

Table 12: Results of using various decoding strategies on the original validation set.

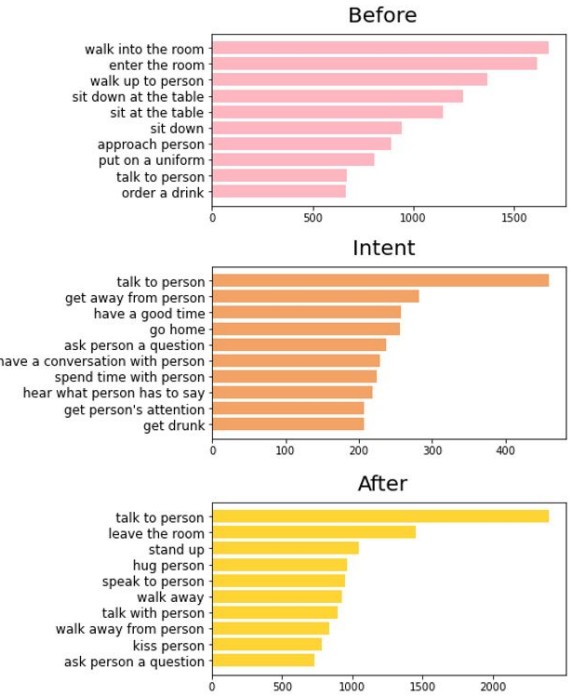

Figure 6: Top 10 inferences and their frequencies most frequently connected to different images for inference type in the training set of VCG

| | DIVE$_{BART}$ | DIVE$_{BLIP}$ |
|---|---|---|
| Backbone | BART-base | BLIP-base |
| Batch Size | 64, 128, **256** | |
| Learning Rate | 1e-5, 2e-5, 3e-5, 4e-5, **5e-5** | |
| $\lambda$ | 0.1, 0.2, 0.3, 0.4, **0.5** | |
| Filtering Threshold | 10 | |
| Dropout rate | 0.3 | 0.1 |
| Epoch | 20 | 10 |
| Training Time (A6000 × 1) | 60h | 50h |

Table 13: Detailed settings for training DIVE framework. Bold text indicates the chosen hyperparameter among we tried.

out most generic inferences from the training set. For each image-inference pair in our training set, 38.8% of pairs have connected to more than one similar image, however, only 18.7% of pairs are connected to more than five similar images. As a result, to make the model see more diverse images in one step, sampling one similar image for one example is the best option.

| | B-2 | C | Entropy | Unique |
|---|---|---|---|---|
| Freezing | 12.86 | 19.07 | 20.93 | 76.05 |
| Full fine-tuning | 12.44 | 18.28 | 20.93 | 75.81 |

Table 14: Results of DIVE$_{BLIP}$ according to freezing visual encoder on original validation set.

| | B-2 | C | Entropy | Unique |
|---|---|---|---|---|
| Freezing | 18.83 | 21.20 | 20.93 | 95.34 |
| Full fine-tuning | 17.34 | 18.49 | 21.00 | 93.63 |

Table 15: Results of DIVE$_{BLIP}$ according to freezing visual encoder on unique validation set.

| | B-2 | C | Entropy | Unique |
|---|---|---|---|---|
| Freezing | 19.04 | 22.52 | 21.23 | 95.48 |
| Full fine-tuning | 16.71 | 18.81 | 21.15 | 94.05 |

Table 16: Results of DIVE$_{BLIP}$ according to freezing visual encoder on novel validation set.

# E Training Setup

In Table 13, we present hyperparameter settings for our models. In addition to that information, we use AdamW (Loshchilov and Hutter, 2019) as our optimizer. We do not adopt any learning rate scheduler or gradient clipping technique. We use six NVIDIA RTX A6000 GPUs, and the whole training procedure can be done within a day. We implement the model code using PyTorch (Paszke et al., 2019) and HuggingFace (Wolf et al., 2020) and we train the model in 16-bit bfloat16 precision for efficiency. We freeze the visual encoders of both DIVE$_{BART}$ and DIVE$_{BLIP}$. In the case of DIVE$_{BLIP}$, our empirical observations suggest that the frozen settings generally produce better performance compared to full fine-tuning settings. As shown in Table 14, 15 and 16, it is possibly due to better resistance to catastrophic forgetting of freezing settings. For DIVE$_{BART}$, we have not tuned the visual encoder (i.e., Faster R-CNN (Ren et al., 2015)), since its discrete sampling procedures pose training difficulties.

To fine-tune DIVE models, we empirically choose for the $t$ and $\lambda$, a filtering threshold and a loss balancing value, respectively. In Table 17, we report the results of DIVE$_{BART}$ with varying

| | B-2 | C | Entropy | Unique |
|---|---|---|---|---|
| $t = 5$ | 12.80 | 19.82 | 21.85 | 81.31 |
| $t = 10$ | 13.33 | 20.26 | 21.09 | 76.09 |
| $t = 20$ | 13.25 | 20.24 | 20.61 | 73.17 |

Table 17: Results of $\text{DIVE}_{BART}$ according to filtering threshold ($t$) on original validation set.

filtering thresholds.

## F   Error Analysis

We provide several representative cases that show the efficacy and limitation of DIVE. In Figure 9 (a), we find that DIVE effectively generates descriptive and plausible inferences by recognizing the facial expressions of people in the image. Moreover, we can also observe that the performance of DIVE can be underestimated by the conventional metrics focusing on n-gram overlaps due to simple human annotations. We further report one of the cases where DIVE generates incorrect inferences, as shown in Figure 9 (b). In this example, although the detailed information such as a cat held by Person2 and the people on the train is explicitly given, DIVE generates some irrelevant inferences to the given context such as "bring the cat to the bathroom". This implies that existing vision language models still lack some commonsense knowledge and reasoning ability to accurately reason over recognized context.

**"engage 1 in conversation"**

Figure 7: Results of various similarity evaluation for generic inference filtering.

Figure 8: Results of various similarity evaluation for generic inference filtering.

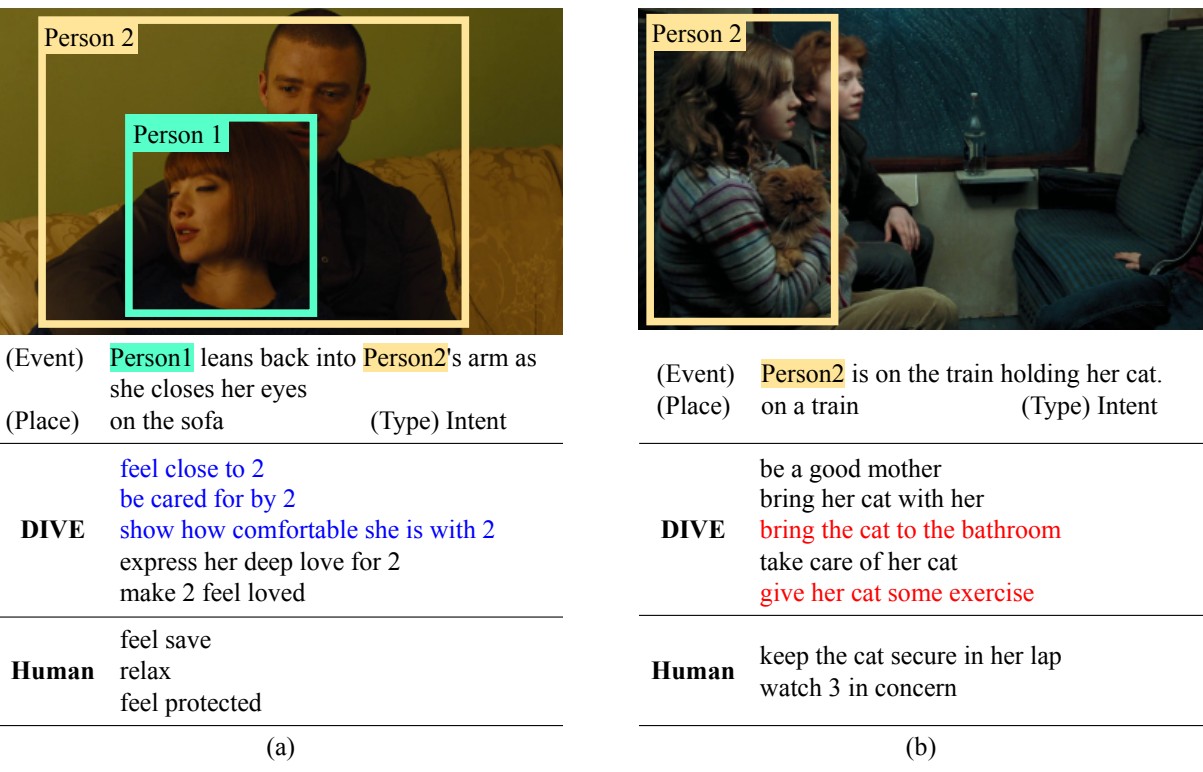

Figure 9: Examples of generated inferences from DIVE on the original VCG validation set. We mark red if the inference is implausible and blue if the inference is both unique and novel.

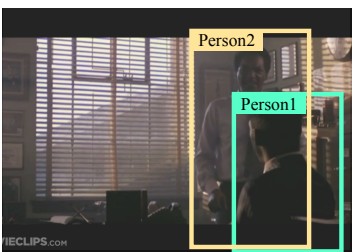

**(Event) Person1 sits calmly in their seat while looking at Person2**

**(Place) in an office**

|  | Before | Intent | After |
|---|---|---|---|
| Human | • be brought into the room for questioning
• sit down
• get called into 2's office
• sit down in front of the dest | • stand their ground
• remain calm | • ask 2 a question
• be interrogated
• get yelled at by 2
• be suspended without pay |
| DIVE | • be escorted to the boss' office by 2
• get asked to sit down by 2
• be called into 2's office
• tell 2 to enter the room
• invite 2 to a conference room | • concentrate on what 2 is saying
• appear calm
• be compliant with 2
• avoid provoking the man in the office
• finish their conversation with 2 | • be escorted out by security
• get out of the office
• give 2 order on what to do
• stay silent
• be threatened by 2 |
| KM-BART | • ask 2 a question
• sit in the chair
• get up from the desk
• find her assigned seat
• sit down in their office chair | • answer questions
• have a conversation
• ask 2 a question
• ask for something
• avoid causing problems | • talk about the incoming threat
• leave the room
• provide security for 2
• have an argument with 2
• continue speaking to 2 |
| VisualCOMET | • lean towards 2
• witness a gunfight
• do something wrong to make 2 aware of it
• hear something that made them upset
• be approached by 2 | • have 2 tell them they are going to leave
• ponder the situation
• understand why they're there
• hear what 2 has to say
• get answer from 2 | • talk about things with 2
• sip on a beverage
• run away
• argue with 2
• stand up |

(a)

**(Event) Person1 has gun in his right hand as he approaches a car**

**(Place) on a highway**

|  | Before | Intent | After |
|---|---|---|---|
| Human | • train to fire the weapon
• lock and load the weapon
• load his gun
• have his car break down | • take 2's car
• forece 2 out of the car | • forcus on the target
• fire a warning shot into the air
• car jack a civilian
• drive off |
| DIVE | • get out of the car with the gun
• lift the gun up
• pull the gun out
• load the gun with ammunition
• pull the gun from its holster | • be ready to shoot
• feel like a big shot
• secure the area
• get something from the car
• make sure the driver was secured | • admire the car in the distance
• knock on the car window
• take someone into custody
• wait for a target
• continue to pursue the car |
| KM-BART | • have the intention of going into the car
• walk out onto the sidewalk
• get out of the car
• exit a building
• see the car | • enter the car
• rob the place
• make sure the car is locked up
• pick up passengers
• back up his partner who is in the car | • talk to 2
• get in the car
• head into the building
• get in the car
• secure the perimeter |
| VisualCOMET | • know where he was going
• sneak out
• be asked to go to the car
• arrive at the scene of the crime
• get out of the car | • drive 2 home
• wants to get 2's attention
• stop 2 from going in the car
• chase after 2
• get 2 to get out of the car | • drive away
• yell for the hostage taker to stop
• enter the car
• arrest the person he's aimed at
• get into the car |

(b)

Figure 10: Comparison of generation examples from DIVE, KM-BART (Xing et al., 2021), VisualCOMET (Park et al., 2020), and human annotations in VCG validation set. We mark red if the inference is implausible and blue if the inference is both unique and novel.