# OpenReview forum: "DIVE: Towards Descriptive and Diverse Visual Commonsense Generation"
_EMNLP/2023/Conference — EMNLP 2023 Main_

### Official Review · Reviewer_TWj1 · 2023-08-02

**Soundness:** 4

**Excitement:**

2: Mediocre: This paper makes marginal contributions (vs non-contemporaneous work), so I would rather not see it in the conference.

**Paper Topic And Main Contributions:**

This paper aims to improve the diversity of vision commonsense generation, by first filtering generic descriptions to build a more balanced training dataset, then adding an extra contrastive objective to encourage the similarity of paired image-text representations.
Experimental results show that the proposed method is effective regarding descriptiveness ( mainly measured by length, distinct n-grams) and generalization quality (measured by traditional n-gram-based metrics such as BLEU and SPICE).


**Reasons To Accept:**

- Improving the diversity and description of vision commonsense generation is an interesting topic.



**Reasons To Reject:**



- The novelty of improving generation diversity by adjusting the training distribution and using contrastive learning to improve consistency is limited. Besides, why contrastive learning can improve generation diversity is not clearly stated.

- The adopted metrics are NOT a good indicator for the claim "descriptive and diverse inferences by capturing important, specific, and detailed information within a visual scene", therefore whether the proposal really captures the specific visual commonsense knowledge, or just learns to produce more unique n-grams, remains unclear.

- The diversity of conditional text generation is well-investigated, and experimental results can be improved by incorporating more comparisons, such as unlikelihood training and contrastive decoding.


**Reproducibility:**

4: Could mostly reproduce the results, but there may be some variation because of sample variance or minor variations in their interpretation of the protocol or method.

**Reviewer Confidence:**

4: Quite sure. I tried to check the important points carefully. It's unlikely, though conceivable, that I missed something that should affect my ratings.

---

> ### Author Rebuttal · Authors · 2023-08-29
>
> We sincerely appreciate the thoughtful and valuable comments. We have endeavored to address all concerns in our response.
>
> **Reasons to Reject 1. Limited novelty and unclear reasons for diversity improvements by contrastive learning**
>
> A. We present an unprecedented filtering and learning methodology that utilizes the structure and semantics of visual commonsense graphs to improve descriptiveness and diversity in visual commonsense generation. We would like to highlight the following points.
>
> - Our filtering method is anchored on the principle of semantic concentration of images — a pioneering approach to identifying generic inferences. This is in contrast to related work, which has largely focused on data augmentation [1] or modifying objective functions [2].
>
> - We propose a novel contrastive retrieval learning method, uniquely leveraging the structural information from visual commonsense graphs. This method enforces the model to identify of specific differences among similar images, based on the structural information from visual commonsense graphs.
>
> - Indeed, our contrastive learning method exhibits significant performance improvements on filtered datasets, where contrastive sets are constructed from more semantically concentrated images, possibly leading to a more precise understanding of semantics.
>
> **Reasons to Reject 2. Inappropriate adopted metrics**
>
> A. We adopt the metrics that reflect important aspects of descriptive and diverse inferences, which are widely employed to evaluate the descriptiveness and diversity of generated texts [3, 4, 5, 6]. Furthermore, our human evaluations confirm that the proposed methods indeed capture specific visual information, leading to improvements in plausibility, descriptiveness, and diversity. If the reviewer can suggest additional metrics that would be more effective in measuring the descriptiveness and diversity of our generated inferences, we would be most grateful for such recommendations.
>
> **Reasons to Reject 3. Comparisons with unlikelihood training and contrastive decoding**
>
> A. We will investigate other training and decoding methods in the future. In this work, we have focused on mitigating the data skewness of visual commonsense resources towards generic inferences. Therefore, we have mainly compared our models with those trained on full data or augmented data, while keeping the original training and decoding method.
>
>
> [1] Liu et al., Generating diverse and descriptive image captions using visual paraphrases, CVPR 2019.
>
> [2] Luo et al., Discriminability objective for training descriptive captions, CVPR 2018.
>
> [3] Liu et al., Generating Diverse and Descriptive Image Captions Using Visual Paraphrases, ICCV 2019.
>
> [4] Park et al., VisualCOMET: Reasoning about the Dynamic Context of a Still Image, ECCV 2020.
>
> [5] Xing et al., KM-BART: Knowledge Enhanced Multimodal BART for Visual Commonsense Generation, ACL-IJCNLP 2021.
>
> [6] Mou et al., Sequence to Backward and Forward Sequences: A Content-Introducing Approach to Generative Short-Text Conversation, COLING 2016.

---

### Official Review · Reviewer_3SLs · 2023-08-12

**Soundness:** 3

**Excitement:**

3: Ambivalent: It has merits (e.g., it reports state-of-the-art results, the idea is nice), but there are key weaknesses (e.g., it describes incremental work), and it can significantly benefit from another round of revision. However, I won't object to accepting it if my co-reviewers champion it.

**Paper Topic And Main Contributions:**

The paper proposed a framework, named DIVE. DIVE improves the scriptiveness and diversity in visual commonsense generation by generic inference filtering and contrastive retrieval learning.

**Reasons To Accept:**

1. The authors pay attention to scriptiveness and diversity in the visual commonsense generation, which are important aspects in this task.
2. Writing is good and easy to understand.
3. Experiment results and cases are abundant.

**Reasons To Reject:**

While the task is fairly novel, the method is simple. Most of the deep learning achievements in recent years have not been used.

**Reproducibility:**

3: Could reproduce the results with some difficulty. The settings of parameters are underspecified or subjectively determined; the training/evaluation data are not widely available.

**Reviewer Confidence:**

2: Willing to defend my evaluation, but it is fairly likely that I missed some details, didn't understand some central points, or can't be sure about the novelty of the work.

---

> ### Author Rebuttal · Authors · 2023-08-29
>
> We sincerely appreciate the thoughtful and valuable comments. We have endeavored to address all concerns in our response.
>
> **Reasons To Reject 1. Novel tasks but simple methods**
>
> A. We believe that our methodology is both novel and carefully designed. Throughout our research, we have explored a wide range of deep learning models and techniques, and chosen the optimal ones. We would like to highlight the following points.
>
> - We have introduced innovative training frameworks that integrate distinct filtering and training strategies. Notably, our filtering method is anchored on the principle of semantic concentration of images — a pioneering approach to identifying generic inferences. Moreover, we employ a contrastive retrieval learning method, uniquely leveraging the structural information from visual commonsense graphs.
>
> - We have considered various recent vision models including CLIP [1], ViT [2], BLIP [3], and Champagne [4] to quantify semantic concentration. Based on our qualitative evaluation, CLIP outperforms the rest in gauging the similarity.
>
> - We have considered several training techniques including direct fine-tuning and reinforcement learning, widely used in recent commonsense work [5, 6], for retrieval learning. However, direct fine-tuning shows performance degradation of about 10%, while reinforcement learning methods show considerable training instability.
>
> - Finally, our methodology outperforms contemporary visual commonsense generation models in terms of descriptiveness and diversity, as well as generation quality, which are substantiated through our extensive automatic and human evaluations.
>
> We are always seeking to improve our methodology. If the reviewer can recommend any recent developments in deep learning that may be relevant to our research, we would greatly appreciate the guidance.
>
> [1] Radford et al., Learning transferable visual models from natural language supervision, ICML 2021.
>
> [2] Dosovitskiy et al., An Image is Worth 16x16 Words: Transformers for Image Recognition at Scale, ICLR 2021.
>
> [3] Li et al., BLIP: Bootstrapping Language-Image Pre-training for Unified Vision-Language Understanding and Generation, ICML 2022.
>
> [4] Han et al., CHAMPAGNE: Learning Real-world Conversation from Large-Scale Web Videos, ICCV 2023.
>
> [5] Liu et al., Rainier: Reinforced Knowledge Introspector for Commonsense Question Answering, ACL 2022.
>
> [6] Yu et al., Fusing Pre-trained Language Models with Multimodal Prompts through Reinforcement Learning, CVPR 2023.

---

### Official Review · Reviewer_kR3u · 2023-08-12

**Soundness:** 4

**Excitement:**

4: Strong: This paper deepens the understanding of some phenomenon or lowers the barriers to an existing research direction.

**Paper Topic And Main Contributions:**

The paper presents a new framework, namely DIVE, which aims to enhance the generation of descriptive and diverse commonsense descriptions based on given images. DIVE consists of two components: data cleaning, and a contrastive training method, which can be plugged into any existing model. The dataset in this task contains many generic descriptions that could potentially limit a model's learning capability. To overcome this problem, the paper introduces a measure that is contingent on the similarity of related images for a description to discard poor-quality instances from the training data. Subsequently, DIVE employs an InfoNCE loss to encourage models to generate descriptive and diverse text. Through empirical evaluation, the paper illustrates DIVE can improve existing models in various scenarios (Tables 1-6). Also, a human assessment, conducted by crowdworkers, confirms the superiority of DIVE over baselines. The paper also provides a qualitative analysis of generated descriptions to highlight where DIVE might generate less plausible text.

**Reasons To Accept:**

1. The focus of the paper on commonsense in multimodal settings can be of interest to the community.
2. The heuristic proposed in the paper, aimed at filtering out generic descriptions, underscores the crucial role of data quality.
3. The paper is well-written and provides thorough experiments with compelling results.


**Reasons To Reject:**

Overall, the paper presents an intuitive method with sufficient empirical evidence to support its effectiveness. I don’t find any major reason that would stop me from recommending acceptance. Nevertheless, I do have the following minor concerns:

1. The reliance on an additional model (CLIP as mentioned in the paper) for deriving feature representations to filter the dataset needs to be explicitly highlighted in the Introduction.

2. No actual results were reported in Section 5.1, although the findings are described.


**Reproducibility:**

4: Could mostly reproduce the results, but there may be some variation because of sample variance or minor variations in their interpretation of the protocol or method.

**Reviewer Confidence:**

3: Pretty sure, but there's a chance I missed something. Although I have a good feel for this area in general, I did not carefully check the paper's details, e.g., the math, experimental design, or novelty.

**Typos Grammar Style And Presentation Improvements:**

I found Appendix E very interesting and hope the authors can make room for it in the main body.

---

> ### Author Rebuttal · Authors · 2023-08-29
>
> We sincerely appreciate the thoughtful and valuable comments. We have endeavored to address all concerns in our response.
>
> **Reasons To Reject 1. Need to highlight the reliance on an additional model**
>
> A. We agree with your comment. We will explicitly highlight the usage of CLIP for filtering in the Introduction section of the camera-ready version.
>
> **Reasons To Reject 2. Absence of reported results in Section 5.1**
>
> A. We appreciate the feedback. Section 5.1 is the discussion of the results shown in Table 8. We will clarify it in the camera-ready version.
>
> **Improvements 1. Consideration for including Appendix E in the main text.**
>
> A. We appreciate your suggestion. We will incorporate the Appendix E section into the main body by utilizing the additional page given on acceptance.

---

### Official Review · Reviewer_Ukok · 2023-08-12

**Soundness:** 4

**Excitement:**

4: Strong: This paper deepens the understanding of some phenomenon or lowers the barriers to an existing research direction.

**Missing References:**

A suggestion, rather than a missing reference: the current method seemingly filters generic inferences quite aggressively. Table 14 in the Appendix shows that tweaking the filtering threshold $t$ doesn’t impact the results too significantly; however, it may be worth considering whether keeping more generic inferences around would be beneficial --- in Dataset Cartography (Swayamdipta et al, 2022), "easy" inputs are shown to still have a specific/unique value during training.

**Paper Topic And Main Contributions:**

The paper proposes a new framework for visual commonsense generation that yields more diverse and descriptive outputs. The framework consists of two stages: generic inference filtering, and contrastive retrieval learning. They are complementary and improve the quality of the generated inferences, according to several metrics as well as a human evaluation.

The contributions of the paper are the new framework, as well as an extensive set of experiments and an analysis of the same.

**Questions For The Authors:**

1. L297: the other images $h_k$ are treated as negative. Isn’t it possible for any of them to be valid for the image $h_p$ as well? How would this be addressed?
2. L445: 17.3% on which metric? (mention absolute points, not relative)
3. Sec 3.1: the goal is to determine which images are “semantically concentrated” (L223). Wouldn’t the similarity metric defined be heavily influenced by visual similarity of images rather than semantic similarity, especially when using CLIP embeddings? I’d like the paper to contain an explanation of why this wouldn’t be the case, along with corresponding proof (even if in the appendix).
    1. Follow-up: Don’t the images have a text description as well? Why wouldn’t you use them instead of / in addition to the images to gauge semantic similarity?
4. Why do you freeze the visual features of DIVE-BLIP and not DIVE-BART? I may be missing something here because the DIVE-BART architecture is never actually discussed (listed in Reasons to Reject 1.5).
5. A brief description of word-level entropy in Sec 5.2 (even if in the appendix) would be helpful.
6. Why do you use greedy decoding in Table 14 instead of nucleus sampling, as in the rest of the paper? Are the results similar?


**Reasons To Accept:**

1. The proposed framework improves both BLIP and a vision-language enhanced BART significantly across many metrics, both automatic (Tables 1-3) and human (Table 7). This shows the merit of the framework itself, which could potentially continue to be used as the underlying VL models improve.
2. The experimentation is neat and thorough. The authors conduct evaluations using various metrics as well as a human evaluation, to confirm that the metrics used mirror human judgment. The appendix contains further experiments (although plugs in the main text would be useful). The ablation analysis is also well discussed.
3. There (seemingly) has not been a large body of work in visual commonsense generation over the past few years. However, as VCG methods improve, visual commonsense reasoning shows more potential to improve. These generation methods could be used to create (perhaps slightly noisy) datasets at large scales, which could be used to train VL models on visual commonsense reasoning; or to create more challenging visual commonsense reasoning benchmarks for VL models (after human filtering).
4. The paper is well written and easy to understand. The authors attach code.

**Reasons To Reject:**

1. More information about the methodology and experiments is needed.
    1. L289: “we construct a set of _similar_ images H”. How is similar defined?
    2. L301-303: “representations of an image and a text from a vision-language model”. Which vision-language model? Where in the model are the representations are extracted from?
    3. Eq 5: The loss equation is incorrect: the original loss in Park et al 2020 calculates loss over the length of a string. Here, the loss is seemingly calculated over the number of inferences, where each is conditioned on the previous inferences (not the previous tokens of the string). This is probably a typo, but should definitely be corrected.
    4. Eq 6: For each image $h_p$ in a batch, the authors construct $H$ and calculate the loss according to this equation. However, there are likely multiple $s_p$s for any $h_p$. Is the loss summed over all $s_p$s?
    5. Very importantly, how are GPT-2 and BART adapted to support a vision input? Especially BART, which is used in these experiments as DIVE-BART – although KM-BART is cited, a brief discussion is definitely needed here (more than L362-379) so the readers know what the model looks like.
2. A random sample of outputs would be appreciated, even if in the Appendix.


**Reproducibility:**

5: Could easily reproduce the results.

**Reviewer Confidence:**

4: Quite sure. I tried to check the important points carefully. It's unlikely, though conceivable, that I missed something that should affect my ratings.

**Typos Grammar Style And Presentation Improvements:**

The variable $t$ is defined as the reasoning type (L233) as well as the filtering threshold (Eq 2).

---

> ### Author Rebuttal · Authors · 2023-08-29
>
> We sincerely appreciate the thoughtful and valuable comments. We have endeavored to address all concerns in our response.
>
> **Reasons To Reject 1. More information about the methodology and experiments**
>
> A. We will elaborate on the methodology and experiments in the camera-ready version.
>
> **- L289: Definition of similar images $H$**
>
> - We define “similar images” as those that share the same inference result. For instance, given the subgraph of Figure 2(a), three sets of similar images can be constructed: one set includes three images that share “See what 1 is holding”, another set comprises two images that share “Get a closer look at 1”, and the third set contains multiple images that share “Speak to 1”, as the inference results.
>
> **- L301-303: Details of extracting representations from a vision-language model**
>
> - We extract the representations using BART and BLIP models, which are simultaneously trained via contrastive retrieval learning. We input images and their corresponding events and places to the models' encoder, and their inference results to the models' decoder. Image representations $V_h$ and text representations $T_s$ are subsequently extracted by averaging the output feature vectors from the final layers of the model’s encoder and decoder, respectively.
>
> **- Eq 5: Incorrect loss equation**
>
> - Thank you for the correction. For an image $h \in H$, we identify its corresponding ground-truth inference $s = [w_1, w_2, …, w_k]$ as a sequence of tokens. The loss is then computed conditioned on previous tokens in this sequence.
>
> **- Eq 6: Loss computation with multiple ground-truth inference results $s_p$s for any image $h_p$**
>
> - We randomly select one $s_p$ for the loss calculation if multiple $s_p$s are associated with an image $h_p$.
>
> **- Adaptation of GPT-2 and BART to support a vision input**
>
> - We use projected Region of Interest (RoI) feature representations of an image from the pre-trained Faster R-CNN [1] as a vision input to the language models. For each image, Faster R-CNN detects several objects and generates their bounding boxes and classification results as RoI features. We extract the feature representations fed into the final classification layer of Faster R-CNN. The extracted representations are subsequently projected to fit the embedding dimension of the language models.
>
> **Reasons To Reject 2. A random sample of outputs**
>
> A. Thank you for the comment. In addition to the reported samples, we will further report random samples of outputs in the Appendix of the camera-ready version. This [Link](https://ibb.co/M81TRXw) illustrates two randomly sampled outputs from DIVE and the baselines.
>
> **Question 1. L297: Negative images that are potentially valid**
>
> A. We apply the contrastive retrieval loss only when a single unique positive image is present. If there are multiple positive images, we exclude the contrastive retrieval loss from our training objective.
>
> **Question 2. L445: 17.3% on which metric**
>
> A. In Table 6, DIVE_BART achieves a SPICE score of 8.36, which represents an improvement of 1.23 points (17.3%) over KM-BART’s score of 7.13.
>
> **Question 3. Influences by visual similarity when using CLIP embeddings and reasons not to use text descriptions**
>
> A. We have verified that our similarity metric is not over-influenced by visual similarity. Upon analyzing images categorized by our metric as similar and dissimilar, we have not identified any distinct visual patterns that distinguish them. However, we have observed that solely relying on texts or combining them with images can lead to over-influence by overlapping words and text lengths, resulting in leaving irrelevant images or filtering out relevant, descriptive images. We will involve the similarity evaluation results and the comparison among the outcomes of image-only, text-only, and combined image-text settings of similarity evaluation in the camera-ready version. The [Link](https://ibb.co/VHbZFnV) illustrates comparisons of top-2 similar and dissimilar images by the similarity measurement strategies on two randomly sampled examples.
>
> **Question 4. Reasons to freeze the visual features of DIVE-BLIP and not DIVE-BART**
>
> A. We have frozen the visual encoders of both DIVE-BLIP and DIVE-BART. For DIVE-BLIP, our empirical observations indicate that freezing settings achieve better performance on average than the full fine-tuning settings, as shown in the following tables, possibly due to better resistance to catastrophic forgetting of freezing settings. For DIVE-BART, we have not tuned the visual encoder (i.e., Faster RCNN), since its discrete sampling procedures pose training difficulties. We will detail the training settings more comprehensively in the camera-ready version.
>
> - Evaluations on the original VCG validation set
>
> |     DIVE-BLIP    |     BLEU-2    |     METEOR    |     CIDER    |     SPICE    |     Unique    |     Novel    |     Entropy    |
> |---|---|---|---|---|---|---|---|
> |     Freezing    |     12.86    |     11.39    |     19.07    |     6.90    |     76.05    |     56.50    |     20.93    |
> |     Full fine-tuning    |     12.44    |     11.16    |     18.28    |     6.51    |     75.81    |     57.21    |     20.94    |
>
> - Evaluations on the unique VCG validation set
>
> |     DIVE-BLIP    |     BLEU-2    |     METEOR    |     CIDER    |     SPICE    |     Unique    |     Novel    |     Entropy    |
> |---|---|---|---|---|---|---|---|
> |      Freezing    |     18.83    |     12.52    |     21.20    |     8.08    |     95.34    |     57.33    |     20.93    |
> |     Full fine-tuning    |     17.34    |     11.98    |     18.49    |     7.01    |     93.63    |     58.03    |     21.00    |
>
> - Evaluations on the novel VCG validation set
>
> |     DIVE-BLIP    |     BLEU-2    |     METEOR    |     CIDER    |     SPICE    |     Unique    |     Novel    |     Entropy    |
> |---|---|---|---|---|---|---|---|
> |      Freezing    |     19.04    |     12.23    |     22.52    |     8.36    |     95.48    |     59.69    |     21.23    |
> |     Full fine-tuning    |     16.71    |     11.47    |     18.81    |     7.19    |     94.05    |     59.93    |     21.15    |
>
> **Question 5. Description of word-level entropy**
>
> A. We will elaborate on the word-level entropy, as well as the other metrics, in the camera-ready version. Word-level entropy measures the average log probability of the words in a generated inference. We use the unigram probability of a word, which refers to the frequency of the word divided by the total number of words in the training set.
>
> **Question 6. Greedy decoding in Table 14**
>
> A. We use greedy decoding in Table 14 because it shows the performance trend among different threshold values more distinctly. We also observe a similar trend using nucleus sampling. We provide the results of nucleus sampling on the original VCG validation set here, which will be reported in the camera-ready version.
>
> |     DIVE-BART    |     BLEU-2    |     METEOR    |     CIDER    |     SPICE    |     Unique    |     Novel    |     Entropy    |
> |------------------|---------------|---------------|--------------|--------------|---------------|--------------|----------------|
> |     t=5          |     12.80     |     11.47     |     19.82    |     7.39     |     81.31     |     60.70    |     21.85      |
> |     t=10         |     13.33     |     11.48     |     20.26    |     7.33     |     76.09     |     54.20    |     21.09      |
> |     t=20         |     13.24     |     11.33     |     20.24    |     7.10     |     73.17     |     51.58    |     20.61      |
>
> **Suggestion 1. Consideration of whether keeping more generic inferences around would be beneficial**
>
> A. We appreciate your suggestion. We will examine the impact of the amount of generic inferences in the future.
>
> **Improvements 1. Variable $t$ is defined in both L233 and Eq. 2.**
>
> A. Thank you for the correction. We will denote the reasoning type as $r$.
>
> [1] Ren et al., Faster R-CNN: Towards Real-Time Object Detection with Region Proposal Networks, NIPS 2015.

---

### Official Review · Reviewer_k6e2 · 2023-08-13

**Soundness:** 4

**Excitement:**

3: Ambivalent: It has merits (e.g., it reports state-of-the-art results, the idea is nice), but there are key weaknesses (e.g., it describes incremental work), and it can significantly benefit from another round of revision. However, I won't object to accepting it if my co-reviewers champion it.

**Paper Topic And Main Contributions:**

This work is addressing the issue of generating more descriptive and diverse inferences in visual commonsense generation tasks. Towards this, the paper proposes a framework called DIVE with 1) generic inference filtering method to balance the Visual Commonsense Graphs (VCG) dataset by removing high frequency and low semantic images, and 2) contrastive retrieval learning method to maximize the agreement during training. Evaluated on VCG, the framework performs SOTA results in terms of descriptiveness and diversity.

**Reasons To Accept:**

The authors provide insights on how to generate descriptive and diverse commonsense inferences about visual scenes as the first exploration.

The authors develop the framework to address generation issues on the skewed visual commonsense resources with comparable results. Overall the work is well-written and easy to follow. The method and experiment design are convincing.

**Reasons To Reject:**

For the `descriptive and diverse` visual commonsense generation tasks, using data after filtering out the generic ones as the first step of solution seems not surprising. The ablation study results (Table 8) also show some more drops without generic inference filtering than without contrastive retrieval learning.

**Reproducibility:**

5: Could easily reproduce the results.

**Reviewer Confidence:**

3: Pretty sure, but there's a chance I missed something. Although I have a good feel for this area in general, I did not carefully check the paper's details, e.g., the math, experimental design, or novelty.

**Typos Grammar Style And Presentation Improvements:**

A. It would be great to make Figure 2 more clear with more illustrations on the images and text (e.g. events and tasks).

B. Is the annotation of Person1 and Person2 reversed in Figure 5a?

---

> ### Author Rebuttal · Authors · 2023-08-29
>
> We sincerely appreciate the thoughtful and valuable comments. We have endeavored to address all concerns in our response.
>
> **Reasons To Reject 1. Surprisingness of the filtering method.**
>
> A. We believe that our data filtering method has produced innovative results, since it is challenging to achieve actual improvements in descriptiveness and diversity through filtering methods. As evidence, simply filtering out the most frequent inferences leads to less descriptive generation in our human evaluation (49.2%-win rate vs. KM-BART). Furthermore, the removal of inferences generally leads to a noticeable degradation in generation quality due to the reduced training data. By focusing on the semantic concentration of images, our filtering method effectively identifies and removes generic inferences, thereby improving descriptiveness and diversity while minimizing generation quality degradation. We will clarify this point in the camera-ready version.
>
> **Improvement 1. Clarity of Figure 2.**
>
> A. We appreciate your feedback on this. We will incorporate additional illustrations of images accompanied by textual descriptions of events and tasks, in Figure 2 of the camera-ready version.
>
> **Improvement 2. Annotations in Figure 5a.**
>
> A. We appreciate your correction. In Figure 5a, the annotations of *Person 1* and *Person 2* are mistakenly swapped. This will be corrected in the camera-ready version.

---

### Meta-Review · Area_Chair_a4nz · 2023-09-19

**Recommendation:** 5

**Metareview:**

The paper introduces a novel framework for generating visual commonsense content, resulting in more diverse and detailed outputs. The experimentation is meticulously conducted, and the ablation analysis is well-explained. The proposed framework brings substantial improvements to both BLIP and a vision-language enhanced BART across various metrics, including both automated (Tables 1-3) and human evaluation (Table 7).

The reviewers find the paper well-written and clear, and they appreciate the innovative and robust approach. Following the rebuttal phase, all reviewers have voted in favor of accepting the paper. I recommend accepting the paper.

---

### Decision · Program_Chairs · 2023-10-07

**Decision:**

Accept-Main

**Comment:**

The paper introduces a novel framework for generating visual commonsense content, resulting in more diverse and detailed outputs. The experimentation is meticulously conducted, and the ablation analysis is well-explained. The proposed framework brings substantial improvements to both BLIP and a vision-language enhanced BART across various metrics, including both automated (Tables 1-3) and human evaluation (Table 7).

The reviewers find the paper well-written and clear, and they appreciate the innovative and robust approach. Following the rebuttal phase, all reviewers have voted in favor of accepting the paper. I recommend accepting the paper.